# Validation of Corporate Probability of Default Models Considering Alternative Use Cases

**Michael Jacobs, Jr.**

Wholesale 1st Line Model Development Validation Services, PNC Financial Services Group—Balance Sheet Analytics & Modeling/Model Development, 340 Madison Avenue, New York, NY 10022, USA; michael.jacobsjr@pnc.com; Tel.: +1-917-324-2098

**Abstract:** In this study, we consider the construction of *through-the-cycle* ("TTC") PD models designed for credit underwriting uses and *point-in-time* ("PIT") PD models suitable for early warning uses, considering which validation elements should be emphasized in each case. We build PD models using a long history of large corporate firms sourced from Moody's, with a large number of financial, equity market and macroeconomic variables as candidate explanatory variables. We construct a Merton model-style *distance-to-default* ("DTD") measure and build hybrid structural reduced-form models to compare with the financial ratio and macroeconomic variable-only models. In the hybrid models, the financial and macroeconomic explanatory variables still enter significantly and improve the predictive accuracy of the TTC models, which generally lag behind the PIT models in that performance measure. We conclude that care must be taken to judiciously choose the manner in which we validate TTC vs. PIT models, as criteria may be rather different and be apart from standards such as discriminatory power. This study contributes to the literature by providing expert guidance to credit risk modeling, model validation and supervisory practitioners in controlling the model risk associated with such modeling efforts.

**Keywords:** probability of default; point-in-time; through-the-cycle; credit risk; model validation; credit underwriting; early warning systems; regulatory capital; model risk

**JEL Classification:** G21; G28; M40; E47

## 1. Introduction

It is expected that financial market participants have accurate measures of a counterparty's capacity to fulfil future debt obligations, conventionally measured by a credit rating or a score, typically associated with a *probability of default* ("PD"). Most extant risk rating methodologies distinguish model outputs considered *point-in-time* ("PIT") vs. *through-the-cycle* ("TTC"). Although these terminologies are widely used in the credit risk modeling community, there is some confusion about what these terms precisely mean. In our view, based upon first-hand experience in this domain and a comprehensive literature review, at present a generally accepted definition for these concepts remains elusive, apart from two points of common understanding. First, PIT PD models should leverage all available information, borrower-specific and macroeconomic, which most accurately reflect default risk at any point of time. Second, TTC PD models abstract from cyclical effects and measure credit risk over a longer time period encompassing a mix of economic conditions, exhibiting "stability" of ratings wherein dramatic changes are related mainly to fundamental and not transient economic fluctuations. However, in reality this distinction is not so well defined, as idiosyncratic factors can influence systematic conditions (e.g., credit contagion), and macroeconomic conditions can influence obligors' fundamental creditworthiness.

There is an understanding in the industry of what distinguishes PIT and TTC constructs, typically defined by how PD estimates behave with respect to the business cycle. However, how this degree of "TTC-ness" vs. "PIT-ness" is defined varies considerably

across institutions and applications, and there is no consensus around what thresholds should be established for certain metrics, such as measures of ratings volatility. As a result, most institutions characterize their rating systems as "Hybrid". While this may be a reasonable description, as arguably the TTC and PIT constructs are ideals, this argument fails to justify the use cases of a PD model where there may be expectations that the model is closer to either one of these poles.

In this study, we develop empirical models that avoid formal definitions of PIT and TTC PDs, rather deriving constructs based upon common sense criteria prevalent in the industry, illustrating which validation techniques are applicable to these approaches. Based upon this empirical approach, we characterize PIT and TTC credit risk measures and discuss the key differences between both rating philosophies. In the process, we address the validation of PD models under both rating philosophies, highlighting that the validation of either system exhibits a particular set of challenges. In the case of the TTC PD rating models, in addition to flexibility in determining measurement of the cycle, there are unsettled questions around the rating stability metric thresholds. In the case of PIT PD rating models, there is the additional question of demonstrating the accuracy of PD estimates at the borrower level, which may not be obvious from observing average PD estimates versus default rates over time. Finally, considering both types of model, there is the question of whether the relative contributions of risk factors are conceptually intuitive, as we would expect that certain variables would dominate in either of these constructs.

There are some additional comments in order to motivate this research. First, there is a misguided perception in the literature and industry that PIT models contain only macroeconomic factors, and that TTC models contain only financial ratios, whereas from a modeling perspective there are other dimensions that define this distinction that we elaborate upon in this research. Furthermore, it may be argued that the validation of a TTC or PIT PD model involves assessing the validity of the cyclical factor, which if not available to the validator, may be accounted for only implicitly. One possibility is for the underlying cycle to be estimated from historical data based upon some theoretical framework, but in this study, we prefer commonly used macroeconomic factors in conjunction with obligor-level default data, in line with industry practice. Related to this point, we do not explicitly address how TTC PD models can be transformed into PIT PD rating models, or vice versa. While the advantage of such alternative constructs is that it can be validated based upon an assumption regarding the systematic factor using the methodologies applicable to each type of PD model, we prefer to validate each as specifically appropriate. The rationale for our approach is that the alternative runs the risk of introducing significant model and estimation risk, thereby leading to the validity of such validation being rendered questionable as compared to testing a pure PIT or TTC PD model.

We employ a long history of borrower level data sourced from Moody's, around 200,000 quarterly observations from a large population of rated larger corporate borrowers (at least USD 1 billion in sales and domiciled in the U.S. or Canada), spanning the period from 1990 to 2015. The dataset comprises an extensive set of financial ratios, macroeconomic[1] and equity market variables as candidate explanatory variables. We build a set of PIT models with a 1-year default horizon and macroeconomic variables, and a set of TTC models with a 3-year default horizon and only financial ratio risk factors.

The position of this research in the academic literature is at the intersection of two streams of inquiry. First, there are a series of empirical studies that focus on the factors that determine corporate default and the forecasting of this phenomenon, which include Altman (1968) and Duffie and Singleton (1999). At the other end of the spectrum, there are mainly theoretical studies that focus on modeling frameworks for either understanding corporate default (e.g., Merton (1974)), or else for perspectives on the TTC vs. PIT dichotomy (e.g., Kiff et al. 2004; Aguais et al. 2008; Cesaroni 2015). In this paper, we blend these considerations of theory and empirics, while also addressing the prediction of default and TTC/PIT construct.

We would like to emphasize that we believe the principal contribution of this paper to be mainly in the domain of practical application rather than methodological innovation. Many practitioners, especially in the wholesale credit and banking book space, still use the techniques employed in this paper. We see our contribution as proposing a structured approach to constructing a suite of TTC and PIT models, while combining reduced form and structural modeling aspects, and then by further proposing a framework for model validation. We would note that many financial institutions in this space do not have such a framework. For example, a lot of banks are still using TTC Basel models that are modified for PIT uses, such as stress testing or portfolio management. Furthermore, a preponderance of banks in this space do not employ hybrid financial and Merton-style models for credit underwriting. In sum, our contribution transcends the academic literature to address issues relevant to financial institution practitioners in the credit risk modeling space, which we believe uniquely positions this research.

The summary of our empirical results are as follows. We present the leading two models in each class of PIT and TTC design, both having favorable rank ordering power, intuitive relative weights on explanatory variables and rating mobility metrics. We also perform predictive accuracy analysis and specification testing, where we observe that the TTC designs are more challenged than the PIT designs in performance, and that unfortunately all designs show some signs of model misspecification. This observation argues for the consideration of alternative risk factors, such as equity market information. In view of this, from the market value of equity and accounting measures of debt for these firms, we are able to construct a Merton model-style *distance-to-default* ("DTD") measure and construct hybrid structural reduced-form models, which we compare with the financial ratio and macroeconomic variable-only models. We show that adding DTD measures to our leading models does not invalidate the variables chosen, significantly augments model performance and in particular increases the obligor-level predictive accuracy of the TTC models.

Finally, let us introduce the remainder of this paper, which will proceed as follows. In Section 2, we review the relevant literature, where we address a survey of PD modeling in general, and then the issues around rating philosophy in particular. In Section 3, we address modeling methodology, which we partition into the domains of econometric modeling and statistical assumptions. Section 4 encompasses the empirical analysis of this study, as a description of the modeling data, estimation and validation results. In Section 5, we conclude and summarize the study, discuss policy implications and provide thoughts on avenues for future research.

## 2. Literature Review

Traditional credit risk models focus on estimating the PD, rather than on the magnitude of potential losses in the event of default (or *loss-given-default*—"LGD"), and typically specify "failure" to be bankruptcy filing, default, or liquidation, thereby ignoring consideration of the downgrades and upgrades in credit quality that are measured in *mark-to-market* ("MTM") credit models. Such *default mode* ("DM") models estimate credit losses resulting from default events only, whereas MTM models classify any change in credit quality as a credit event. There are three broad categories of traditional models used to estimate PD: expert systems, including artificial neural networks; rating systems; and credit scoring models.

The most commonly used traditional credit risk measurement methodology is the PD scoring model. The seminal model in this domain is the *multiple discriminant analysis* ("MDA") of Altman (1968). Mester (1997) documents the widespread use of credit scoring models amongst banks, with 97% and 70% of them using them to approve credit card and small business loan applications, respectively. Credit scoring models are relatively inexpensive to implement and do not suffer from the subjectivity and inconsistency of expert systems. The spread of these models throughout the world was first surveyed by Altman and Narayanan (1997). The authors find that it is not so much the models'

differences across countries of diverse sizes and in various stages of development that stands out, but rather their similarities. A popularly used vended PD scoring model in the industry is the private firm model of *Moody's Analytics* ("MA"; Dwyer et al. 2004).

Merton (1974) models equity in a levered firm as a call option on the firm's assets with a strike price equal to the debt repayment amount. The PD is determined by valuing the call option using an iterative method to estimate the unobserved variables that determine this, the market value of assets and the volatility of assets, combined with the amount of debt liabilities that have to be repaid at a given credit horizon in order to calculate the firm's *distance-to-default* ("DTD"). DTD is the number of standard deviations between the current asset values and the debt repayment amount, so the higher it is, the lower the PD. In an important example of this, in the *CreditEdge$^{TM}$* ("CE") public firm model of MA, an empirical PD using historical default experience is estimated using a historical database of default rates to determine an empirical estimate of the PD, denoted the *expected default frequency* ("EDF"). As CE EDF scores are obtained from equity prices, they are more sensitive to changing financial circumstances than external credit ratings that rely predominately on credit underwriting data.

Modern methods of credit risk measurement can be traced to two alternative branches in the asset pricing literature of academic finance. In contrast to the option of the theoretic structural approach, which was pioneered by Merton (1974), a *reduced form approach* utilizing *intensity-based models* to estimate stochastic *hazard rates* follows a study pioneered by Jarrow and Turnbull (1995) and Duffie and Singleton (1999). These two schools of thought offer differing methodologies to accomplish the central task of all credit risk measurement models, which is the estimation of PDs. The structural approach models the economic process of default, whereas reduced form models decompose risky debt prices in order to estimate the random intensity process underlying default. The proprietary model *Kamakura Risk Manager* ("KRM"), where the econometric approach (the so-called *Jarrow-Chava Model*—"JCM") is a reduced-form model based upon the research of Chava and Jarrow (2004), attempts to explicitly adjust for liquidity effects. However, noise from embedded options and other structural anomalies in the default risk-free market further distorts risky debt prices, thereby impacting the results of intensity-based models.

There are several more recent studies of particular relevance to this research that could be mentioned, but for the sake of brevity, we will refer the reader to the comprehensive literature review of Altman (2018). However, we will highlight one important study from a methodological perspective by Jiang (2021). This paper investigates the incentive of credit rating agencies to bias ratings using a semiparametric, ordered-response model. Using Moody's rating data from 2001 to 2016, the author finds that firms related to Moody's shareholders were more likely to receive better ratings.

In the recent literature on PD modeling, there has been a proliferation of studies investigating machine learning techniques. Kim (2005) applies *adaptive learning networks* (ALN), which is a nonparametric model, on both financial and non-financial variables to predict S&P credit ratings. Yu et al. (2008) proposes a six stage neural network ensemble learning model to assess a credit risk measurement on Japanese consumer credit card application approval and UK corporations. Khashman (2010) investigates three neural networks based on a back propagation learning algorithm on the *German Credit Approval* dataset. The architecture of these neural networks is different according to various parameters used in the model, such as hidden units, learning coefficients, momentum rate and random initial weight range. Pacelli and Azzollini (2011) provide an overview of different types of neural networks used in the credit-rating literature. Among all artificial intelligence techniques, *support vector machines* ("SVMs") have demonstrated powerful classification abilities (Cortes and Vapnik 1995; Kim and Sohn 2010; Vapnik 2013; Xiao et al. 2016). Khandani et al. (2010) studies the general *classification and regression tree technique* ("CART") on a combination of traditional credit factors and consumer banking transactions to predict consumer credit risk. Veronezi (2016) applied *random forest* ("RF") and *multilayer perceptron* ("MLP") techniques to predict corporate credit ratings using their financial data. Finally, in

addition to all these frequently used techniques, some researchers study other approaches to provide a credit scoring model. Peng et al. (2011) introduce three *multiple criteria decision making* ("MCDM") methods to evaluate classification algorithms for financial risk prediction. Chen (2012) investigates the *rough set theory* ("RST") approach to classify Asian banks' ratings. Finally, some researchers take one step further and integrate multiple techniques to achieve a higher accuracy, such as Yeh et al. (2012), who combine random forest feature selection with different approaches such as RST and SVM.

One of the key motivations behind the new generation of PD models being developed in the industry, as well as in this research, is to provide a suite of models that can accommodate multiple uses, such as TTC models for credit underwriting or *risk weighted assets* ("RWA"), as well as PIT models for credit portfolio management or early warning. One point to highlight is that despite the growing literature on TTC credit ratings, there is still no consensus on the precise definition of this concept, except the general agreement that TTC ratings are adjusted to not reflect cyclical effects. The Basel guidelines (BIS 2006) describe a PIT rating system as a construct that uses all currently available obligor-specific and aggregate information to estimate an obligor's PD, in contrast to a TTC rating system that, while using obligor-specific information, tends not to adjust ratings in response to changes in macroeconomic conditions. However, the types of such cyclical effects and how they are measured differ considerably in the literature as well as in practice.

First, a number of studies have come up with a formal definition of the concepts of PIT and TTC PD estimates and rating systems. These include Loeffler (2004), who explores the TTC methodology in a structural credit risk model based on Merton (1974), in which a firm's asset value is separated into a permanent and a cyclical component. In this model, TTC credit ratings are based on forecasting the future asset value of a firm under a *stress scenario* for the cyclical component. Kiff et al. (2004) investigate the TTC approach in a structural credit risk model in which the definition of TTC ratings follows the one applied by Hamilton et al. (2011), emphasizing that while anecdotal evidence from credit rating agencies confirm their use of the TTC approach, it turns out that there is no single and simple definition of what a TTC rating actually means. In contrast to the majority of studies in the literature that define PIT and TTC credit measures on the basis of a decomposition of credit risk into idiosyncratic and systematic risk factors, Aguais et al. (2008) follow a frequency decomposition view in which a firm's credit measure is split up into a long-term credit quality trend and a cyclical component which are filtered from the firm's original credit measure by using a smoothing technique based on the filter in Hodrick and Prescott (1997). Furthermore, the authors argue that in the existing literature, there has been little discussion about whether the C in TTC refers to the business cycle or the credit cycle and highlight that these cycles differ considerably from each other regarding their length. They describe a practical framework for banks to compute PIT and TTC PDs through converting PIT PDs into TTC PDs based on sector-specific credit cycle adjustments to the DTD credit measures of the Merton (1974) model derived from a credit rating agency's rating or MA's CE model. Furthermore, they qualitatively discuss key components of PIT-TTC default rating systems and how these systems can be implemented in banks. On the other hand, Cesaroni (2015) analyzes PIT and TTC default probabilities of large credit portfolios in a Merton single-factor model, where the author defines the TTC PD as the expected PIT PD, where the expectation is taken over all possible states of a systematic risk factor. Repullo et al. (2010) propose translating PIT PDs into TTC PDs by ex post smoothing the estimated PIT PDs with countercyclical scaling factors. In connection with the industry next-generation PD model redevelopment efforts and this research, with the objective of supporting TTC vs. PIT ratings, these results support not having formal definitions of TTC vs. PIT ratings, in light of the diversity of approaches seen in the literature.

Second, several studies analyze the ratings of major rating agencies regarding their PIT vs. TTC orientation. These include the Altman and Rijken (2004) who find, based on credit scoring models, that major credit rating agencies pursue a long-term view when assigning ratings, putting less weight on short-term default indicators and hence indicating

TTC orientation. Loeffler (2013) shows for Standard and Poor's and Moody's rating data that these agencies have a policy of changing a rating only if it is unlikely to be reversed in the future and argues that this can explain the empirical finding that rating changes lag changes of an obligor's default risk, consistent with the general view of TTC ratings. Altman and Rijken (2006) analyze the TTC methodology of rating agencies from an investor's PIT perspective and quantify the effects of this methodology on the objectives of rating stability, rating timeliness, and performance in predicting defaults. Among other results, they find that TTC rating procedures delay migration in agency ratings, on average, by $\frac{1}{2}$ a year on the downgrade side and $\frac{3}{4}$ of a year on the upgrade side, and that from the perspective of an investor's one-year horizon, TTC ratings significantly reduce the short-term predictive power for defaults. Several papers, such as Amato and Furfine (2004) and Topp and Perl (2010), analyze actual rating data and show that these ratings vary with the business cycle, even though these ratings are supposed to be TTC according to the policies of the credit rating agencies. Loeffler (2013) estimates long-run trends in market-based measures of one-year PDs using different filtering techniques. They show that agency ratings contribute to the identification of these long-run trends, thus providing evidence that credit rating agencies follow to some extent a TTC rating philosophy. To summarize, many studies find that the ratings of major rating agencies show both PIT as well as TTC characteristics, which is consistent with the notion of hybrid rating systems. In connection with this research and industry redevelopment efforts, with the objective of supporting TTC vs. PIT ratings, these results support not having "hard" mobility metric thresholds in evaluating the model output.

Third, the rating philosophy is important from a regulatory and supervisory perspective, as well as from a credit underwriting perspective, not least because capital requirements for banks and insurance firms depend upon credit risk measures. Studies that discuss TTC PDs in the context of Basel II or as a remedy for the potential pro-cyclical nature of Basel II (BIS 2006) include Repullo et al. (2010), who compare smoothing the input of the Basel II formula by using TTC PDs or smoothing its output with a multiplier based on GDP growth. They prefer the GDP growth multiplier because TTC PDs are worse in terms of simplicity, transparency, cost of implementation, and consistency with banks' risk pricing and risk management systems. Cyclicality of credit risk measures also plays an important role in the context of Basel III (BIS 2011), which states that institutions should have sound internal standards for situations where realized default rates deviate significantly from estimated PDs, and that these standards should take account of business cycles and similar systematic variability in default experience. In two separate consultation papers issued in 2016, The European Banking Authority (2016) proposes to explicitly leave the selection of the rating philosophy to the banks, whereas the *Basel Committee for Banking Supervision* ("BCBS"; BIS 2016—"Bank for International Settlements") proposes requiring banks to follow a TTC approach to reduce the variability in PDs and thus RWAs across banks.

Finally, the rating philosophy should influence the validation of rating systems, but the challenges to validate TTC models have been largely ignored in the literature. The BCBS (BIS 2005) further stresses that in order to evaluate the accuracy of PDs reported by banks supervisors need to adapt their PD validation techniques to the specific types of banks' credit rating systems, in particular with respect to their PIT vs. TTC orientation. However, methods to validate rating systems have paid very little attention to the rating philosophy or focused on PIT models. For example, Cesaroni (2015) observes that predicted default rates are PIT, and thus the validation of a rating system "should" operate on PIT PDs from a theoretical perspective. Petrov and Rubtsov (2016) explicitly mention that they have not yet developed a validation framework consistent with their PIT/TTC methodology.

To conclude this section, we mention an important paper on PIT PD modeling by Đurović (2019) where a framework is proposed for retail PD modeling in accordance with the *International Reporting Financial Standards 9* accounting regulation. The model is based upon a term structure of PD conditional to the given forward-looking macroeconomic

dynamics. Due to data limitations, a key impediment in forward-looking modelling, the author proposes and illustrates a model averaging technique for the quantification of macroeconomic effects on the PD.

## 3. Methodology

In this section, we outline the econometric technique and statistical PD modeling in the industry. In principle, for classification tasks including default prediction, while one could use the same loss functions as those used for regression (i.e., the *ordinary least squares* criterion; "OLS") in order to optimize the design of the classifier, this would not be the most reasonable way to approach such problems. This is because in classification, the target variable is discrete in nature; hence, alternative measures to those employed in regression are more appropriate for quantifying the quality of model fit. This discussion could be motivated by the classification problem for default prediction through *Bayesian decision theory*, which has conceptual simplicity and aligns well with common sense and possesses a strong optimality flavor with respect to the probability of an error in classification. However, given that the focus and contribution of this paper does not lie in the domain of econometric technique, we will defer such discussion and focus on the *logistic regression model* ("LRM") technique, as it is widely employed and well understood in the literature and practice.

Considering the 2-class $\{\omega_i\}_{i=1}^2$ case for the LRM that is relevant to PD modeling, the first step is to express the *log-odds* (or the *logit function*) of the posterior probabilities as a linear function of the risk factors:

$$\ln\left(\frac{P(\omega_1|\mathbf{x})}{P(\omega_2|\mathbf{x})}\right) = \theta^T\mathbf{x}, \tag{1}$$

where $x = (x_1,..,x_k) \in \mathbb{R}^k$ is a $k$ dimensional feature vector and $\theta = (\theta_1,..,\theta_k) \in \mathbb{R}^k$ is a vector of coefficients, and we define $x_1 = 1$ so that the intercept is subsumed into $\theta$. In that, $P(\omega_1|\mathbf{x}) + P(\omega_2|\mathbf{x}) = 1$:

$$P(\omega_1|\mathbf{x}) = \frac{1}{1 + \exp\left(-\theta^T\mathbf{x}\right)} = \sigma\left(-\theta^T\mathbf{x}\right), \tag{2}$$

where the function $\sigma\left(-\theta^T\mathbf{x}\right)$ is known as the *logistic sigmoid* (or *sigmoid link*) and has the mathematical properties of a cumulative distribution function that ranges between 0 and 1, with a domain on the real line. Intuitively, this can be viewed as the conditional PD of a score $\theta^T\mathbf{x}$ where higher values indicate greater default risk.

We may estimate the parameter vector $\theta$ by the method of *maximum likelihood estimation* ("MLE") given a set of training samples, with observations of explanatory variables $\{\mathbf{x}_n\}_{n=1}^N$ and binary dependent variables $\{y_n\}_{n=1}^N$, where $y_n \in \{0,1\}$. The likelihood function is given by:

$$P(y_1,\ldots,y_N|\theta) = \prod_{n=1}^N \left(\sigma\left(-\theta^T\mathbf{x}_n\right)\right)^{y_n}\left(1 - \sigma\left(-\theta^T\mathbf{x}_n\right)\right)^{1-y_n}. \tag{3}$$

The practice is to consider the negative *log-likelihood function* (or the *cross-entropy error*), a monotonically increasing transformation of (3), for the purposes of computational convenience:

$$L(\theta) = -\sum_{n=1}^N y_n \ln\left(\sigma\left(-\theta^T\mathbf{x}_n\right)\right) + (1 - y_n)\ln\left(1 - \sigma\left(-\theta^T\mathbf{x}_n\right)\right). \tag{4}$$

Equation (4) is *minimized* with respect to $\theta$ using iterative methods, such as the steepest descent of Newton's scheme.

We note an important property of this model that is computationally convenient and leads to stable estimation under most circumstances. Since $\sigma\left(-\theta^T \mathbf{x}_n\right) \in (0,1)$ according to the properties of the sigmoid link function, it follows that the variance-covariance matrix $\mathbf{R}$ is positive definite, which implies that the Hessian matrix $\nabla^2 L(\theta)$ is positive definite. In turn, this implies that the negative log-likelihood function $L(\theta)$ is *convex*, and as such this guarantees the existence of a unique minimum to this optimization. However, maximizing the likelihood function may be problematic in the case where the development dataset is *linearly separable*. In such a case, any point on the hyperplane $\hat{\theta}_{MLE}^T \mathbf{x} = 0$ (out of an infinite number of such hyperplanes) that solves the classification task and separates the training samples in each class does so *perfectly*, which means that every training point is assigned a posterior probability of class membership equal to one (or $\sigma\left(\hat{\theta}_{MLE}^T \mathbf{x}\right) = \frac{1}{2}$).

In this case, the MLE procedure forces the parameter estimate to be infinite ($\hat{\theta}_{MLE}^T \to \infty$), which means geometrically that the sigmoid link function approaches a step function and not an s-curve as a function of the score. This is basically a case of overfitting the development sample, which can be controlled by techniques such as k-fold cross-validation, or including a regularization term inside a corresponding cost function that controls the magnitudes of the parameter estimates (e.g., LASSO techniques for a linear penalty function $C(\theta|\lambda) = \lambda\|\theta\|$ with a cost parameter $\lambda$).

We conclude this section by discussing the statistical assumptions underlying the LRM model. Logistic regression does not make many of the key assumptions of OLS regression regarding linearity, normality of error terms, homoscedasticity of the error variance and the measurement level. Firstly, LRM does not assume a linearity relationship between the dependent variable and estimator[2], which implies that we can accommodate non-linear relationships between the independent and dependent variables without non-linear transformations of the former (although we may choose to do so for other reasons, such as treating outliers), which yields more parsimonious and more intuitive models. Another way to look at this is since we are applying the log-odds transformation to posterior probabilities, by construction we have a linear relationship in the risk drivers and do not necessarily require additional transformations. Secondly, the independent variables do not need to be multivariate normal, which equivalently means that the error terms need not be multivariate normal either. While there is an argument that if the error terms are actually multivariate normal (which is probably not true in practice), then imposing this assumption leads to efficiency gains and possibly a more stable solution; at the same time, there are many more parameters to be estimated. That is because in the normal case we not only have to estimate the $k$ regression coefficients $\theta = (\theta_1, .., \theta_k) \in \mathbb{R}^k$, but we also have to estimate the entire variance-covariance matrix (i.e., the variance-covariance matrix in the LRM is a function of $\theta$), which is $O\left(\frac{k^2}{2}\right)$ additional operations and could lead to a more unstable model depending upon data availability as well as more computational overhead. Thirdly, since the variance-covariance matrix also depends on $\mathbf{x}$ by construction through the sigmoid link function, variances need not be homoscedastic for each level of the independent variables (while if we imposed a normality assumption, we would require this assumption to hold as well). Lastly, the LRM can handle ordinal and nominal independent variables as they need not be metric (i.e., interval or ratio scaled), which leads to more flexibility in model construction and again avoids counterintuitive transformations and more parameters to be estimated.

However, some other assumptions still apply in the LRM setting. First, the LRM requires the dependent variable to be binary, while other approaches (e.g., *ordinal logistic regression*—"OLR" or the *multinomial regression model*—"MRM") allow the dependent variable to be polytomous, which implies more granularity in modeling. This is because reducing an ordinal or even metric variable to a dichotomous level loses a lot of information, which makes this test inferior compared to OLR in these cases. In the case of PD modeling, if credit states other than default are relevant (e.g., significant downgrade short of default, or prepayment), then this could result in biased estimates and mismeasurement

of default risk. However, we note in this regard that for many portfolios, data limitations (especially for large corporate or commercial and industrial portfolios) prevents application of OLR for more states than default (e.g., prepayment events may not be identifiable in data), and conceptually we may argue that observations of ratings have elements of expert judgment and are not "true" events (although in wholesale, the definition of default is partly subjective). An assumption related to this is the *independence of irrelevant alternatives*, which states that relative odds of a binary outcome should not depend on other possible outcomes under consideration. In the statistics and econometrics literature, there is debate not only about how critical this assumption is, but also on ways to test this assumption and the value of such tests (Cheng and Long 2006; Fry and Harris 1996; Hausman and McFadden 1984; Small and Hsiao 1985).

Another important assumption is that the LRM requires the observations to be independent, which means that that the data points should not come from any *dependent samples design* (e.g., matched pairings or panel data.) While obviously that is not completely the case in PD modeling in that we have dependent observations, in practice this may not be a very material violation, since if we are capturing most or all of the relevant factors influencing default, then anything else is likely to be idiosyncratic (especially if we are including macroeconomic factors). While in this implementation we are not assuming a parametric distribution for the error terms in the LRM, there are still certain properties that the errors should exhibit, in order for us to have some assurance that the model is not grossly mis-specified (e.g., symmetry around zero, lack of outliers.) However, there is some debate in the literature on the criticality of this assumption, as well as the best way to evaluate LRM residuals (Li and Shepherd 2012; Liu and Zhang 2017).

Finally, we conclude this section by a discussion of the model methodology within the empirical context. The modeling approach as outlined in this section, and the model selection process as elaborated upon in subsequent sections, is common to both PIT and TTC constructs. However, we impose the constraint that only financial factors are considered in the TTC construct, while both the former and macroeconomic variables are considered for the PIT models. This is in addition to the difference in default horizon and other model selection criteria, which results in a differentiation in the TTC and PIT outcomes, in terms of rating mobility and relative factor weights considered intuitive in each construct—i.e., high (lower) rating mobility, and greater (lower) weight on shorter (longer) term financial factors for the PIT (TTC) models.

## 4. Empirical Analysis

### 4.1. Description of Modeling Data

The following data are also used for the development of the models in this study:

- Compustat[TM]: Standardized fundamental and market data for publicly traded companies including financial statement line items and industry classifications (*Global Industry Classification Standards*—"GICS" and *North American Industry Classification System*—"NAICS") over multiple economic cycles from 1979 onward. These data include default types such as bankruptcy, liquidation, and rating agency's default rating, all of which are part of the industry standard default definitions.
- Moody's Default Risk Service[TM] ("DRS") Rating History: An extensive database of rating migrations, default and recovery rates across geographies, regions, industries, and sectors.
- Bankruptcydata.com: A service provided by New Generation Research, Inc. ("NGR") providing information on corporate bankruptcies.
- The Center for Research in Security Prices[TM] ("CRSP") U.S. Stock Databases: This product is comprised of a database of historical daily and monthly market and corporate action data for over 32,000 active and inactive securities with primary listings on the NYSE, NYSE American, NASDAQ, NYSE Arca and Bats exchanges and include CRSP broad market indexes.

A series of filters are applied to this Moody's population to construct a population that is closely aligned with the U.S.'s large corporate segment of companies that are publicly rated and have publicly traded equity. In order to construct and achieve this using Moody's data, the following combination of NAICS and GICS industry codes, region and historical yearly Net Sales are used:

1. Non-*commercial and industrial* ("C&I") obligors defined by the following NAICS codes below, are not included in the population:

   - Financials
   - *Real Estate Investment Trust* ("REIT" or *Real Estate Operating Company* ("REOC")
   - Government
   - Dealer Finance
   - Not-for-Profit, including museums, zoos, hospital sites, religious organizations, charities, and education

2. A similar filter is performed according to GICS (see below) classification:

   - Education
   - Financials
   - Real Estate

3. Only obligors based in the U.S. and Canada are included.
4. Only obligors with maximum historical yearly Net Sales of at least USD 1B are included.
5. There are exclusions for obligors with missing GICS codes, and for modeling purposes obligors are categorized into different industry segments on this basis.
6. Records prior to 1Q91 are excluded, the rationale being that capital markets and accounting rules were different before the 1990s, and the macroeconomic data used in the model development are only available after 1990. As one-year change transformations are amongst those applied to the macroeconomic variables, this cutoff is advanced a year from 1990 to 1991.
7. Records that are too close to a default event are not included in the development dataset, which is an industry standard approach, the rationale being that the records of an obligor in this time window do not provide information about future defaults of the obligor, but more likely the existing problems that the obligor is experiencing. Furthermore, a more effective practice is to base this on data that are 6–18 (rather than 1–12) months prior to the default date, as this typically reflects the range of timing between when statements are issued and when ratings are updated (i.e., usually it takes up to six months, depending on time to complete financials, receive them, input, and complete/finalize the ratings).
8. In general, the defaulted obligors' financial statements after the default date are not included in the modeling dataset. However, in some cases, obligors may exit a default state or "cure" (e.g., emerge from bankruptcy), in which cases, only the statements between default date and cured date are not included.

In our opinion, these data exclusions are reasonable and in line with industry standards, sufficiently documented and supported and do not compromise the integrity of the modeling dataset.

The time periods considered for the Moody's data is the development period Q191–Q415. Shown in Table 1 below is the comparison of the modeling population by GICS industry sectors, where for each sector defaulted obligors columns represent the percent of defaulted obligors in the sector out of entire population. The data are concentrated in Consumer Discretionary (20%), Industrials (17%), Tech Hardware and Communications (12%), and Energy except E&P (11%).

**Table 1.** Large Corporate Modeling Data—GICS Industry Segment Composition for All Moody's Obligors vs. Defaulted Moody's Obligors (1991–2015).

| GICS Industry Segment | All Moody's Obligors | Defaulted Moody's Obligors |
|---|---|---|
| Consumer Discretionary | 19.6% | 30.9% |
| Consumer Staples | 8.4% | 6.4% |
| Energy | 7.6% | 5.9% |
| Healthcare Equipment and Services | 2.9% | 2.9% |
| Industrials | 31.6% | 15.1% |
| Materials | 10.5% | 11.3% |
| Pharmaceuticals and Biotechnology | 2.7% | 0.2% |
| Software and IT Services | 2.5% | 1.8% |
| Technology Hardware and Communications | 4.3% | 11.3% |
| Utilities | 7.6% | 5.6% |

A similar industry composition is shown below in Table 2 according to the NAICS classification system.

**Table 2.** Large Corporate Modeling Data—NAICS Industry Segment Composition for All Moody's Obligors vs. Defaulted Moody's Obligors (1991–2015).

| NAICS Industry Segment | All Moody's Obligors | Defaulted Moody's Obligors |
|---|---|---|
| Agriculture, Forestry, Hunting and Fishing | 0.2% | 0.4% |
| Accommodation and Food Services | 2.3% | 2.9% |
| Waste Management % Remediation Services | 2.4% | 2.1% |
| Arts, Entertainment and Recreation | 0.7% | 1.0% |
| Construction | 1.7% | 2.5% |
| Educational Services | 0.1% | 0.2% |
| Healthcare and Social Assistance | 1.6% | 1.6% |
| Information Services | 11.5% | 12.1% |
| Management Compensation Enterprises | 0.1% | 0.1% |
| Manufacturing | 37.7% | 34.4% |
| Mining, Oil and Gas | 6.8% | 8.6% |
| Other Services (e.g., Public Administration) | 0.4% | 0.6% |
| Professional, Scientific and Technological Services | 2.3% | 2.5% |
| Real Estate, Rentals and Leasing | 0.9% | 1.6% |
| Retail Trade | 9.6% | 12.4% |
| Transportation and Warehousing | 5.4% | 7.0% |
| Utilities | 8.3% | 5.4 |
| Wholesale Trade | 7.0% | 2.7 |

The model development dataset contains financial ratios and default information that are based upon the most recent data available from DRS$^{TM}$, Compustat$^{TM}$ and bankruptcy-data.com, so that the data are timely and a priori should be give the benefit of the doubt with respect to favorable quality. Furthermore, the model development time period of 1Q91–4Q15 spans two economic downturn periods and complete business cycles, the length of which are another factor supporting a verdict of good quality.

Related to this point, we plot the yearly one- and three-year default rates in the model development dataset, shown below in Figure 1. As the goal of model development is to establish for each risk driver that the preliminary trends observed match that of our expectations, there is sufficient variation in this data to support quantitative methods of parameter estimation, further supporting the suitability of the data from a quality perspective.

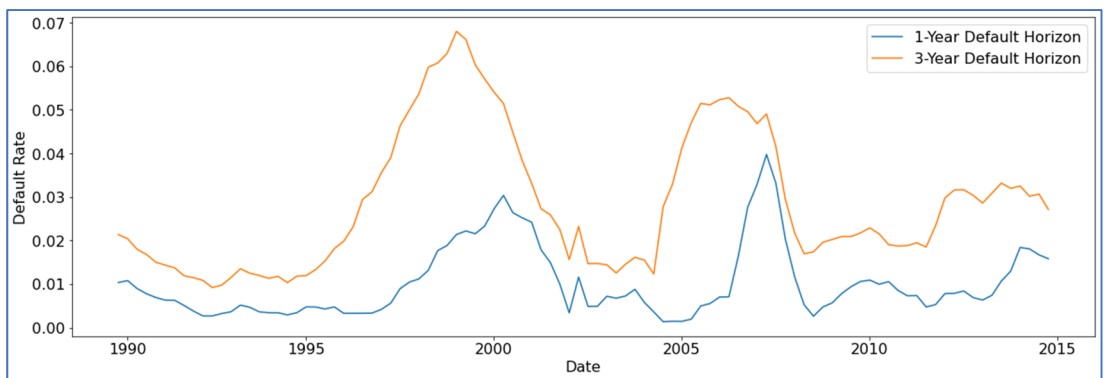

**Figure 1.** Large Corporate Modeling Data—One- and Three-Year Horizon Default Rates over Time (1991–2015).

The following are the categories and names of the explanatory variables appearing in the final candidate models[3]:

- Size: Change in Total Assets ("CTA"), Total Liabilities ("TL")
- Leverage: Total Liabilities to Total Assets Ratio ("TLTAR")
- Coverage Cash Use Ratio ("CUR"), Debt Service Coverage Ratio ("DSCR")
- Efficiency: Net Accounts Receivables Days Ratio ("NARDR")
- Liquidity: Net Quick Ratio ("NQR"), Net Working Capital to Tangible Assets Ratio ("NWCTAR")
- Profitability: Before Tax Profit Margin ("BTPM")
- Macroeconomic" Moody's 500 Equity Price Index Quarterly Average Annual Change ("SP500EPIQAAC"), Consumer Confidence Index Annual Change ("CCIAC")
- Merton Structural: Distance-to-Default ("DTD")

In the subsequent tables (Tables 3–6) we present the summary statistics for the variables that appear in our final models. These final models were chosen based upon an exhaustive search algorithm in conjunction with 5-fold cross-validation, and we have chosen the leading two models in either the PIT and TTC constructs, as well as incorporating the DTD risk factor or not[4]. The counts and statistics vary slightly across models, as the Python libraries that we utilize do not accommodate missing values, but nonetheless the differences in these statistics across models are minimal. The counts of observations vary narrowly from about 150 K to observations of about 165 K. The default rate is consistently about 1% (3%) for the PIT (TTC) models.

**Table 3.** Summary Statistics—Moody's Large Corporate Financial and Macroeconomic Explanatory Variables and Default Indicators: 1-Year PIT Model 1.

| Variable | Count | Mean | Standard Deviation | Minimum | 25th Percentile | Median | 75th Percentile | Maximum |
|---|---|---|---|---|---|---|---|---|
| Default Indicator | | 0.01 | 0.10 | 0.00 | 0.00 | 0.00 | 0.00 | 1.00 |
| Change in Total Assets | | 0.14 | 0.35 | −0.40 | −0.01 | 0.06 | 0.17 | 3.21 |
| Total Liabilities to Total Assets | | 0.60 | 0.23 | 0.12 | 0.45 | 0.59 | 0.71 | 1.53 |
| Cash Use Ratio | | 1.90 | 2.84 | −22.43 | 1.41 | 2.06 | 2.65 | 19.00 |
| Net Accounts Receivables Days | 157,353 | 130.25 | 101.44 | 11.26 | 68.98 | 106.74 | 159.43 | 754.09 |
| Net Quick Ratio | | 0.34 | 1.07 | −0.85 | −0.28 | 0.06 | 0.59 | 6.11 |
| Before Tax Profit Margin | | 5.94 | 21.00 | −146.67 | 1.85 | 7.09 | 12.85 | 48.70 |
| Moody's Equity Price Index | | 1.91 | 6.09 | −27.33 | −0.19 | 2.19 | 5.68 | 12.81 |
| Consumer Confidence Index | | 2.34 | 21.58 | −60.97 | −7.02 | 4.89 | 15.35 | 73.21 |

**Table 4.** Summary Statistics—Moody's Large Corporate Financial, Macroeconomic, Merton/Structural Model Distance-to-Default Proxy Measure Explanatory Variables and Default Indicators: 1-Year PIT Model 2.

| Variable | Count | Mean | Standard Deviation | Minimum | 25th Percentile | Median | 75th Percentile | Maximum |
|---|---|---|---|---|---|---|---|---|
| Default Indicator | | 0.01 | 0.10 | 0.00 | 0.00 | 0.00 | 0.00 | 1.00 |
| Change in Total Assets | | 0.14 | 0.35 | −0.40 | −0.01 | 0.06 | 0.17 | 3.21 |
| Total Liabilities to Total Assets | | 0.60 | 0.23 | 0.12 | 0.45 | 0.60 | 0.71 | 1.53 |
| Cash Use Ratio | | 1.90 | 2.83 | −22.43 | 1.40 | 2.06 | 2.64 | 19.00 |
| Net Quick Ratio | 160,002 | 0.34 | 1.06 | −0.85 | −0.28 | 0.06 | 0.59 | 6.11 |
| Before Tax Profit Margin | | 5.98 | 20.93 | −146.67 | 1.86 | 7.10 | 12.88 | 48.70 |
| Moody's Equity Price Index | | 1.93 | 6.08 | −27.33 | −0.19 | 2.19 | 5.68 | 12.81 |
| Consumer Confidence Index | | 2.37 | 21.56 | −60.97 | −7.02 | 4.89 | 15.35 | 73.21 |
| Distance-to-Default | | 0.20 | 0.43 | −.1.32 | 0.02 | 0.07 | 0.18 | 5.26 |

**Table 5.** Summary Statistics—Moody's Large Corporate Financial and Explanatory Variables and Default Indicators: 3-Year TTC Model 1.

| Variable | Count | Mean | Standard Deviation | Minimum | 25th Percentile | Median | 75th Percentile | Maximum |
|---|---|---|---|---|---|---|---|---|
| Default Indicator | | 0.03 | 0.17 | 0.0 | 0.0 | 0.0 | 0.0 | 1.0 |
| Total Liabilities | | 3640.65 | 6741.93 | 8.86 | 422.60 | 1170.45 | 3374.12 | 41,852.00 |
| Total Liabilities to Total Assets | | 0.62 | 0.22 | 0.12 | 0.49 | 0.61 | 0.72 | 1.53 |
| Debt Service Ratio | 150,064 | 16.44 | 52.82 | −25.07 | 1.74 | 4.09 | 9.80 | 409.64 |
| Net Quick Ratio | | 0.24 | 0.93 | −0.85 | −0.30 | 0.02 | 0.47 | 6.11 |
| Before Tax Profit Margin | | 5.50 | 21.08 | −146.67 | 1.57 | 6.72 | 12.40 | 48.70 |

**Table 6.** Summary Statistics—Moody's Large Corporate Financial and Merton/Structural Model Distance-to-Default Proxy Measure Explanatory Variables and Default Indicators: 3-Year TTC Model 2.

| Variable | Count | Mean | Standard Deviation | Minimum | 25th Percentile | Median | 75th Percentile | Maximum |
|---|---|---|---|---|---|---|---|---|
| Default Indicator | | 0.03 | 0.17 | 0.0 | 0.0 | 0.0 | 0.0 | 1.0 |
| Total Liabilities | | 3640.65 | 6741.93 | 8.86 | 422.60 | 1170.45 | 3374.12 | 41,852.00 |
| Total Liabilities to Total Assets | | 0.62 | 0.22 | 0.12 | 0.49 | 0.61 | 0.72 | 1.53 |
| Debt Service Ratio | 150,064 | 16.44 | 52.82 | −25.07 | 1.74 | 4.09 | 9.80 | 409.64 |
| Net Quick Ratio | | 0.24 | 0.93 | −0.85 | −0.30 | 0.02 | 0.47 | 6.11 |
| Before Tax Profit Margin | | 5.50 | 21.08 | −146.67 | 1.57 | 6.72 | 12.40 | 48.70 |
| Distance-to-Default | | 0.20 | 0.42 | −1.32 | 0.02 | 0.07 | 0.28 | 5.26 |

The *Areas Under the Receiver Operating Characteristic Curve* ("AUC") and missing rates for the explanatory variables are summarized in Table 7 at the end of this section[5]. The univariate AUCs range from 0.6 to 0.8 across risk factors, with some expected deterioration when going from the 1- to 3-year default horizon, which is indicative of strong default rank ordering capability amongst these explanatory variables. The missing rates are generally between 5 and 10%, which is indicative of favorable data quality to support model development.

**Table 7.** Moody's Large Corporate Financial and Macroeconomic Explanatory Variables Areas Under the Receiver Operating Characteristic Curve (AUC) and Missing Rates for 1-Year Default Horizon PIT and 3-Year Default Horizon TTC Default Indicators.

| | | PIT 1-Year Default Horizon | | TTC 3-Year Default Horizon | |
|---|---|---|---|---|---|
| Category | Explanatory Variables | AUC | Missing Rate | AUC | Missing Rate |
| Size | Change in Total Assets | 0.726 | 8.52% | | |
| | Total Liabilities | | | 0.582 | 4.64% |
| Leverage | Total Liabilities to Total Assets Ratio | 0.843 | 4.65% | 0.783 | 4.65% |
| Coverage | Cash Use Ratio | 0.788 | 7.94% | | |
| | Debt Service Coverage Ratio | | | 0.796 | 17.0% |
| Efficiency | Net Accounts Receivables Days Ratio | 0.615 | 8.17% | | |
| Liquidity | Net Quick Ratio | 0.653 | 7.71% | 0.617 | 7.17% |

**Table 7.** *Cont.*

|  |  | PIT 1-Year Default Horizon | | TTC 3-Year Default Horizon | |
|---|---|---|---|---|---|
| Profitability | Before Tax Profit Margin | 0.827 | 2.40% | 0.768 | 2.40% |
| Macroeconomic | Moody's 500 Equity Price Index Quarterly Average Annual Change | 0.603 | 0.00% | | |
|  | Consumer Confidence Index Annual Change | 0.607 | 0.00% | | |
| Merton Structural | Distance-to-Default | 0.730 | 4.65% | 0.669 | 4.65% |

*4.2. Econometric Specifications and Model Validation*

In the subsequent tables we present the estimation results and in-sample performance statistics for our final models.

We shall first discuss general features of the model estimation results. Across models, signs of coefficient estimates are in line with economic intuition, and significant levels are indicative of very precisely estimated parameters. AUC statistics indicate that models have a strong ability to rank order default risk, and while as expected this level of discriminatory power declines somewhat at the longer default horizon, in all cases the levels are in line with favorable performance by industry standards.

Regarding the measures of predictive accuracy, the *Hosmer–Lemeshow* tests ("HL") show that the PIT models fit the data well, while the TTC models fail to do so. However, we observe that when we introduce DTD into the TTC models, predictive accuracy increases markedly, as the *p*-values of the HL statistics increase significantly to the point where there is marginal evidence of adequate fit (i.e., the p-values indicate that the TTC models fail only with significance levels greater than 5%). AIC measures are also much higher in the TTC vs. PIT models, but do decline when the DTD risk factors are introduced, consistent with the HL statistics.

We next discuss general features of the estimation that speak to the TTC or PIT qualities of the models. As expected, the TTC models have much lower *Singular Value Decomposition* ("SVD") rating mobility metrics as compared to the PIT models, in the range of about 30–35% in the former as compared about a 70–80% range in the latter. The relative magnitude of the *factor contribution* ("FC") measures, which quantify the proportion of the total score that is accounted for by an explanatory variable, also support that the models are exhibiting TTC and PIT characteristics. This is because intuitively, we observe that in the TTC models there is greater weight on categories considered more important in credit underwriting (i.e., size, leverage and coverage), whereas in the PIT models this trend is reversed and there is greater emphasis on factors considered more critical to early warning or credit portfolio management (i.e., liquidity, profitability or efficiency).

In Table 8 below, we show the estimation results and in-sample performance measures for PIT Model 1 with both financial and macroeconomic explanatory variables for a 1-year default horizon. FCs are higher on more PIT relevant factors as contrasted to factors considered more salient to TTC constructs. Financial risk factors carry a super-majority of the FC compared to the macroeconomic factors, about 90% in the former as compared to about 10% in the latter, which is a common observation in the industry for PD scorecard models. The model estimation results provide evidence of high discriminatory power, as the AUC is 0.8894. The AIC is 7231.9, which, relative to the TTC models, is indicative of favorable predictive accuracy, which is corroborated by the very high the HL p-value of 0.5945. Finally, the SVD mobility metric 0.7184 supports that this model exhibits PD rating volatility consistent with a PIT model.

**Table 8.** Logistic Regression Estimation Results—Moody's Large Corporate Financial and Macroeconomic Explanatory Variables 1-Year Default Horizon PIT Reduced Form Model 1.

| Explanatory Variable | Parameter Estimate | *p*-Value | Factor Weight | AIC | AUC | HL *p*-Value | Mobility Index |
|---|---|---|---|---|---|---|---|
| Change in Total Assets | −0.4837 | 0.0000 | 0.0455 | | | | |
| Total Liabilities to Total Assets | 2.6170 | 0.0104 | 0.1091 | | | | |
| Cash Use Ratio | −0.0428 | 0.0000 | 0.1545 | | | | |
| Net Accounts Receivables Days Ratio | 0.0005 | 0.0000 | 0.2273 | 7231.00 | 0.8894 | 0.5945 | 0.7184 |
| Net Quick Ratio | −0.4673 | 0.0000 | 0.0909 | | | | |
| Before Tax Profit Margin | −0.0161 | 0.0000 | 0.2736 | | | | |
| Moody's Equity Index Price Index Quarterly Average | −0.0189 | 0.0000 | 0.0759 | | | | |
| Consumer Confidence Index Year-on-Year Change | −0.0099 | 0.0000 | 0.0232 | | | | |

In Table 9 below, we show the estimation results and in-sample performance measures for PIT Model 2 with financial, macroeconomic and the Structural–Merton DTD as explanatory variables for a 1-year default horizon. The results are similar to PIT Model 1 in terms of signs of coefficient estimates, statistical significance and relative FCs on financial and macroeconomic variables. DTD enters the model without any deleterious effects on the statistical significance of financial ratios, although the relative contribution of 0.17 absorbs a fair amount of the other variables' factor weights and eclipses that of the macroeconomic variables. That said, we observe that, collectively, financial and Merton DTD risk factors carry a super-majority of the FC compared to the macroeconomic factors, about 89% in the former as compared to about 11% in the latter, which is a common observation in the industry for PD scorecard models. The model estimation results provide evidence of high discriminatory power as the AUC is 0.8895, which is immaterially lower than then the Model 1 version without DTD. The AIC is 7290.0, which, relative to the TTC models, is indicative of favorable predictive accuracy ad also indicates an improvement in fit as compared to the Model 1 version without the structural model DTD variable, which is corroborated by the very high HL *p*-value of 0.5782. Finally, the SVD mobility metric 0.7616 supports that this model exhibits PD rating volatility consistent with a PIT model, and moreover the addition of the DTD Merton model proxy improves the PIT aspect of this model relative to its Model 1 counterpart which does not have this feature.

**Table 9.** Logistic Regression Estimation Results—Moody's Large Corporate Financial, Macroeconomic and Distance-to-Default Explanatory Variables 1-Year Default Horizon PIT Hybrid Reduced Form/Structural-Merton Model 2.

| Explanatory Variable | Parameter Estimate | *p*-Value | Factor Weight | AIC | AUC | HL *p*-Value | Mobility Index |
|---|---|---|---|---|---|---|---|
| Change in Total Assets | −0.4664 | 0.0000 | 0.0485 | | | | |
| Total Liabilities to Total Assets | 2.5385 | 0.0000 | 0.1165 | | | | |
| Cash Use Ratio | −0.0428 | 0.0000 | 0.1650 | | | | |
| Net Quick Ratio | −0.0169 | 0.0000 | 0.0971 | 7290.00 | 0.8895 | 0.5782 | 0.7617 |
| Before Tax Profit Margin | −0.0169 | 0.0000 | 0.2913 | | | | |
| Moody's Equity Index Price Index Quarterly Average | −0.0186 | 0.0000 | 0.0801 | | | | |
| Consumer Confidence Index Year-on-Year Change | −0.0100 | 0.0000 | 0.0267 | | | | |
| Distance to Default | −0.1913 | 0.0052 | 0.1748 | | | | |

In Table 10 below, we show the estimation results and in-sample performance measures for TTC Model 1 with financial explanatory variables for a 3-year default horizon. The signs of coefficient estimates are intuitive, as all are negative (TL, DSCR, NQR and BTBF), except for TLTAR, which is positive. Parameter estimates are all highly statistically significant. FCs are higher on more TTC relevant factors (i.e., 0.17, 0.31 and 0.23 for TL, TLTAR and DSCR, respectively) as contrasted to the factors considered more salient to PIT constructs (i.e., 0.14 for NQR and BTPM). The model estimation results provide evidence of high discriminatory power, as the AUC is 0.8232, but which as expected is somewhat lower than in the comparable PIT models not containing DTD where they range in the range of 0.88–0.89. The AIC is 17,751.6, which, relative to the comparable PIT models, is indicative

of a rather worse predictive power, which is corroborated by the very low HL P-Value of 0.0039, which rejects the null hypothesis that the model is properly specified with respect to a "saturated model" that perfectly fits the data. Finally, the SVD mobility metric 0.3295 supports that this model exhibits PD rating volatility consistent with a TTC model.

**Table 10.** Logistic Regression Estimation Results—Moody's Large Corporate Financial and Macroeconomic Explanatory Variables 3-Year Default Horizon TTC Reduced Form Model 1.

| Explanatory Variable | Parameter Estimate | *p*-Value | Factor Weight | AIC | AUC | HL *p*-Value | Mobility Index |
|---|---|---|---|---|---|---|---|
| Value of Total Liabilities | $-6.97 \times 10^{-6}$ | 0.0000 | 0.1773 | | | | |
| Total Liabilities to Total Assets | 2.0239 | 0.0030 | 0.3133 | | | | |
| Debt Service Coverage Ratio | $-0.0431$ | 0.0000 | 0.2332 | 17,751.00 | 0.8232 | 0.0039 | 0.3295 |
| Net Quick Ratio | $-0.2412$ | 0.0000 | 0.1372 | | | | |
| Before Tax Profit Margin | $-0.0129$ | 0.0000 | 0.1390 | | | | |

In Table 11 below, we show the estimation results and in-sample performance measures for TTC Model 2 having financial and the Structural-Merton DTD explanatory variables for a 3-year default horizon. The signs of coefficient estimates are intuitive, as all are negative (DSCR, NQR and BTBF), except for TLTAR which is positive, and as expected, DTD has a negative parameter estimate. Parameter estimates are all highly statistically significant. FCs are higher on more TTC-relevant factors (i.e., 0.37 and 0.29 for TLTAR and DSCR, respectively) as contrasted to the factors considered more salient to PIT constructs (i.e., 0.08 and 0.09 for NQR and BTPM, respectively). Note that in this model, adding the DTD explanatory variable results in TL not being statistically significant, and we drop it from this specification; additionally, the FC of DTD is 0.17, so that the financial factors still carry most of the relative weight. The model estimation results provide evidence of high discriminatory power, as the AUC is 0.8226, but which as expected is somewhat lower than in the comparable PIT models containing DTD, where they vary in the range of 0.88–0.89. The AIC is 11,834.6, which relative to the comparable PIT models containing DTD (although this is lower than for TTC model 1, so that DTD improves fir materially), is indicative of a rather worse predictive power, which is corroborated by the somewhat low HL *p*-Value of 0.0973, which rejects the null hypothesis that the model is properly specified with respect to a "saturated model" that perfectly fits the data at the 5% significance level, where we would note that this marginal rejection is an improvement over the comparable TTC version of this model without the Merton DTD variable. Finally, the SVD mobility metric of 0.3539 supports that this model exhibits a PD rating volatility consistent with a TTC model, but we note that the rating volatility measure is somewhat higher than in the comparable TTC model not containing the DTD variable.

**Table 11.** Logistic Regression Estimation Results—Moody's Large Corporate Financial, Macroeconomic and Distance-to-Default Explanatory Variables 3-Year Default Horizon TTC Hybrid Reduced Form/Structural-Merton Model 2.

| Explanatory Variable | Parameter Estimate | *p*-Value | Factor Weight | AIC | AUC | HL *p*-Value | Deviance/ Degrees of Freedom | Pseudo R-Squared | Mobility Index |
|---|---|---|---|---|---|---|---|---|---|
| Total Liabilities to Total Assets | 2.9580 | 0.0000 | 0.3707 | | | | | | |
| Debt Service Coverage Ratio | $-0.0428$ | 0.0000 | 0.2917 | 11,834.00 | 0.8226 | 0.0973 | 0.2365 | 0.1491 | 0.3539 |
| Net Quick Ratio | $-0.2403$ | 0.0000 | 0.0808 | | | | | | |
| Before Tax Profit Margin | $-0.0129$ | 0.0000 | 0.0902 | | | | | | |
| Distance to Default | $-0.1541$ | 0.0000 | 0.1666 | | | | | | |

We conclude this section by comparing our results to other similar studies in potentially different methodological or empirical contexts. Our results are consistent with a

series of empirical studies that focus on the factors that determine corporate default and the forecasting of this phenomenon, (e.g., Altman 1968; Jarrow and Turnbull 1995; Duffie and Singleton 1999), in that we confirm that Merton DTD measures may augment the predictive power of models featuring only financial or macroeconomic factors. Where we innovate in this dimension is in incorporating the TTC vs. PIT constructs as separate models, which addresses this stream of literature (e.g., Kiff et al. 2004; Aguais et al. 2008; Cesaroni 2015), thereby blening these considerations of theory and empirics, while also addressing the prediction of default.

## 5. Conclusions

In this study, we have developed alternative simple and general econometrically estimated PD models of both TTC and PIT designs. We have avoided formal definitions of PIT vs. TTC PDs, and rather derived constructs based upon common sense criteria prevalent in the industry, and in the process have illustrated which validation techniques are applicable to these different approaches. Based upon this empirical approach to modeling, we have characterized PIT and TTC credit risk measures and have discussed the key differences between both rating philosophies. In the process, we have addressed the validation of PD models under both rating philosophies, highlighting that the validation of either rating systems exhibits particular challenges. In the case of the TTC PD rating models, in addition to the flexibility in determining the nature of the cycle underlying and its measurement, we have answered questions around the thresholds for rating stability metrics that are not settled. In the case of PIT PD rating models, we have spoken to questions around the rigorous demonstration that PD estimates are accurately estimated at the borrower level, which may not be obvious from optically observing the degree to which average PD estimates track default rates over time. Considering both TTC and PIT PD models, we have addressed the issue of whether the relative contributions of risk factors are conceptually intuitive, the expectation being that certain variables would dominate in either of these constructs.

We have observed that the validation of a PD TTC or PIT rating model involves assessing the economic validity of the cyclical factor, which if, with respect to the specific modeling methodology, may not be available to the validator, or else may be accounted for only implicitly. One possibility is for the underlying cycle of the PD rating model to be estimated from historical rating and default data based upon some theoretical framework. However, in this study we have chosen to propose commonly used macroeconomic factors in conjunction with obligor-level default data, in line with the industry practice of building such models.

We have highlighted features of PIT vs. TTC model design in our empirical experiment, yet have not explicitly addressed how TTC PD rating models can be transformed into corresponding PIT PD rating models, or vice versa. While the advantage of such a construct is that the latter can then be validated based upon an assumption regarding the systematic factor and validated using the methodologies applicable to each type of PD rating models, we have chosen to validate each as specifically appropriate. The rationale for our approach is that the alternative runs the risk of introducing significant model risk (i.e., if the theoretical model is mis-specified), as well as additional estimation risk (i.e., if the parameter estimates need to be extracted from historical data), thereby leading to the validity of such validation being rendered questionable as compared to testing a pure PIT or TTC PD rating model.

We have employed a long history of borrower-level data sourced from Moody's, around 200,000 quarterly observations from a large population of rated larger corporate borrowers (at least USD 1 billion in sales and domiciled in North America), spanning the period from 1990 to 2015. The dataset comprises an extensive set of financial ratios, as well as macroeconomic variables as candidate explanatory variables. We built a set of PIT models with a 1-year default horizon and macroeconomic variables, and a set of TTC models with a 3-year default horizon and only financial ratio risk factors. We presented

the leading two models in each class of PIT and TTC designs, both having favorable rank ordering power, and propose the leading model based upon the relative weights on explanatory variables (i.e., certain variables are expected to have different relative contributions in TTC vs. PIT constructs), as well as rating mobility metrics (e.g., PIT models are expected to show more responsive ratings and TTC models more stable ratings.) We also performed specification testing, where we observe that the TTC designs are more challenged than the PIT designs in this dimension of performance. The latter observation argues for the consideration of alternative risk factors, such as equity market information. In view of this, from the market value of equity and accounting measures of debt for these firms, we were able to construct a Merton model-style DTD measure and construct hybrid structural-reduced form models, which we compare with the financial ratio and macroeconomic variable-only models.

We showed that adding DTD measures to our leading models does not invalidate the variables chosen, significantly augments model performance and in particular increases the obligor-level predictive accuracy and fit to the data of the TTC models. We also found that while all classes of models have high discriminatory power, the TTC models actually perform better along the dimension of predictive accuracy or fit to the data when we incorporate the DTD risk factor.

There are various implications for model development and validation practice, as well as supervisory policy, which can be gleaned from this study. First, it is a better practice to take into consideration the use case for a PD model in designing the model, from a fitness for purpose perspective. That said, we believe that a balance must be struck, since it would be infeasible to have separate PD models for every single use[6], and what we are arguing for is a parsimonious number of separate designs for major classes that satisfy a set of uses with common requirements. Second, in validating PD models that are designed according to TTC or PIT constructs, in validating such models we should have different emphases on which model performance metrics are scrutinized. In light of these observations and contributions to the literature, we believe that this study provides valuable guidance to model development, model validation and supervisory practitioners. Additionally, we believe that our discourse has contributed to resolving the debates around which class of PD models is best fit for purpose in large corporate credit risk applications, showing evidence that reduced form and Merton structural models can be combined in hybrid frameworks in order to achieve superior performance along the lines of better fit to the data as well as lower measured model risk due to model mis-specification.

Finally, we would like to emphasize that we believe the principal contribution of this paper to be mainly in the domain of practical application rather than methodological innovation. Many practitioners, especially in the wholesale credit and banking book space, still use the techniques employed in this paper. We see our contribution as proposing a structured approach to constructing a suite of TTC and PIT models, while combining reduced form and structural modeling aspects, and then by further proposing a framework for model validation. We would note that many financial institutions in this space do not have such a framework. For example, a lot of banks are still using TTC Basel models that are modified for PIT uses, such as stress testing or portfolio management. Furthermore, a preponderance of banks in this space does not employ hybrid financial and Merton-style models for credit underwriting. In sum, our contribution transcends the academic literature to address issues relevant to financial institution practitioners and prudential supervisors in the credit risk modeling space, which we believe uniquely positions this research.

That said, there are various limitations of this study that should be kept at the front of mind in assessing this contribution. First, there are alternative econometric techniques that we have not considered, such as machine learning models. Second, we have limited our inquiry to a large corporate asset class, and results could differ for other portfolio segments. Third, our framework does not admit the consideration of industry specificity in model specification. Fourth, we have not considered the explicit quantification of model risk in

our model validation framework. Finally, we have not addressed jurisdictions apart from the U.S. or a consideration of geographical effects.

Given the wide relevance and scope of the topics addressed in this study, there is no shortage of fruitful avenues along which we could extend this research. Some proposals include, but are not limited to:

- alternative econometric techniques, such as various classes of machine learning models, including non-parametric alternatives;
- asset classes beyond the large corporate segment, such as small business, real estate or even retail;
- applications to stress testing of credit risk portfolios[7];
- the consideration of industry specificity in model specification;
- the quantification of model risk according to the principle of *relative entropy*;
- different modeling methodologies, such as ratings migration or hazard rate models; and
- datasets in jurisdictions apart from the U.S., or else pooled data encompassing different countries with a consideration of geographical effects.

**Funding:** This research has received no external funding nor any other form of support from any outside parties.

**Institutional Review Board Statement:** This is not applicable as this research did not employ human or animal subjects.

**Informed Consent Statement:** This is not applicable as this research did not employ human subjects.

**Data Availability Statement:** The data used in this study combines proprietary and publicly available sources, therefore the dataset employed is not available.

**Conflicts of Interest:** The author declares no conflict of interest.

## Notes

[1] A key limitation of this construct is that with macroeconomic variables common to all obligors, we are challenged in capturing the cross-sectional variation in the sensitivity to systematic factors across firms. This could be addressed by interaction terms between macroeconomic variables and firm specific factors or industry effects, which can be explored in future research.

[2] Note that linearity *does not* mean that the dependent variable has a linear relationship with the explanatory variables (i.e., we can have non-linear transformations of the latter), but rather that the estimator is a linear function (or weighted average) of the dependent variable, which implies that we can obtain our estimator analytically using linear algebra operations as opposed to iterative techniques such as in the LRM.

[3] All candidate explanatory variables are Winsorized at either the 10th, 5th or 1st percentile levels, at either tail of the sample distribution, in order to mitigate the influence of outliers or contamination in data, according to a customized algorithm that analyzes the gaps between these percentiles and caps/floors where these are maximal.

[4] Clarifying our model selection process, we balance multiple criteria, both in terms of statistical performance and some qualitative considerations. Firstly, all models have to exhibit the stability of factor selection (where the signs on coefficient estimates are constrained to be economically intuitive) and statistical significance in k-fold cross validation sub-sample estimation. However, this is constrained by the requirement that we have only a single financial factor chosen from each category. Then, the models that meet these criteria are evaluated according to statistical performance metrics such as AIC and AUC, as well as other considerations such as rating mobility and relative factor weights.

[5] The plots are omitted for the sake of brevity and are available upon request.

[6] We have observed in the industry that a typical bank can have a number of applications for its PD models far into the double digits, and it would be infeasible to have completely separately developed PD models for all such applications.

[7] Refer to Jacobs et al. (2015) and Jacobs (2020) for studies that address model validation and model risk quantification methodologies. These studies include supervisory applications such as *comprehensive capital analysis and review* ("CCAR") and *current expected credit loss* ("CECL"), and further feature alternative credit risk model specifications (including machine learning model), macroeconomic scenario generation techniques, as well as the quantification and aggregation of model risk (including the principle of *relative entropy*).

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
