# Peer review of "Validation of Corporate Probability of Default Models Considering Alternative Use Cases"

_ijfs, doi:10.3390/ijfs9040063_

Round 1

Reviewer 1 Report

In this paper, the authors presented two alternative solutions to probability of default (PD) model of corporates, namely through-the-cycle (TTC) and point-in-time (PIT) PD models. The aim of this paper is clear and structure is overall well-organized. My major concerns are the novelty of this paper. My detailed comments are as follows.

  1. The paper focused on the validation of TTC and PIT PD models. However, the two types of model are not new and the methodology used in this paper are worn out. I, therefore, concern about the novelty of this paper. If the author really wants to convince the readers that he/she provides something new in this paper, the contribution of this paper should be clarified in a clear form.
  2. I noticed that the author conducted a comprehensive literature review but I think an important research field was ignored. Due to the rapid growth of machine learning, artificial intelligence methods have been extensively applied in building PD models, whereas the author failed to refer to those literature in literature review despite they report better performance in several recent studies.

3. Non-linear techniques were also not considered in this paper so I wonder if the paper can be further improved by incorporating non-linear techniques.

Author Response

Please find attached my responses and thanks for your feedback. Please refer to the notes addressed to "reviewer 1".

Reviewer 2 Report

The author applies Logistic Regressions to predict the probability of default (PD) for a large set of firm in the short run (i.e., 1-year) and long run (i.e., 3-year). I tend to give the paper a green light if the author can effectively answer my concerns below.

Major Comments

  1. First and foremost, the purpose of the paper isn't clear. In the abstract and introduction, the author elaborated the importance of separating the TTC and PIT models as they serve different purposes (i.e., underwriting vs early warning). At one point (line 88-90), the author claimed that from a modeling perspective, TTC only includes financial ratios whereas PIT only considers macroeconomic variables. However, this is not what the author reported in the result section: All reported models contain a mixture of both set of variables. Besides using different default indicators as the dependent variable, how are the TTC and PIT model different?

A related question is: if the PIT model only uses macro variables that do not vary across firms, how could the model capture the cross-sectional variations in firms?

  1. The majority of the cited studies are older papers. The author can expand the literature review to some of the more recent papers on credit risk modeling. For example, the recent review article by Altman (2018) needs to be included. Methodologically, Jiang (2021) uses a semiparametric method to estimate the rating probability, meaning that the σ-function in equation 3.1.8 can be left unspecified and be determined by the data. Durovic (2019) is another paper on PIT model that the author wants to cite. Loffler (2004) and Altman and Rijken (2006) are two important studies on TTC.

  1. Line 516-519, the author wrote ‘’These final models were chosen based upon an exhaustive search algorithm in conjunction with 5-fold cross-validation, and we have chosen the leading two models in either the PIT and TTC constructs, as well as incorporating the Distance-to-Default construct or not." This statement is rather vague. Based on what criteria are the "leading" models selected? Is the author trying to select the number of inputs in each model? or trying to select a model whose parameters are stable in subsamples?

  1. In my view, the paper's contribution does not lie on the econometric method, so Section 3 (methodology) does not have to be as long as it currently is. Since the outcome variable under study is binary (i.e., default or not), the discussion about conditional probabilities in a multinomial model and the Bayes rule can be dropped completely. Reporting equation 3.1.8 alone would suffice as the logit model is fairly well-known. The discussion of the Gradient vector and Hessian matrix can be dropped as well.

Minor

  1. Related to my previous comment, the paper would be much easier to follow if the author discuss the method within the empirical context. After reporting the logit model, the author should explain which type of variables (macro, firm-specific, or both?) are considered in TTC and TIC, respectively.

  1. The author should cut the number of tables significantly. As for the in sample fits, the most important measures are the RMSE and AUC. The histogram and calibration curve don't make cross-comparison easier.

  1. Certain parts of the equations are not displayed. For example, line 274, 292, and 295.

  1. The equations are numbered discontinuously: why is 3.1.2 followed by 3.2.3?

  1. There appeared to be several referencing errors throughout the paper. See the texts: ‘’Error! Reference source not found".

  1. Is it necessary to cite seven papers from Jacobs in the last footnote?

References

Edward I Altman. Applications of distress prediction models: What have we learned after 50 years from the z-score models? International Journal of Financial Studies, 6(3):70, 2018.

Edward I Altman and Herbert A Rijken. A point-in-time perspective on through-the-cycle ratings. Financial Analysts Journal, 62(1):54-70, 2006.

Andrija Durovic. Macroeconomic approach to point in time probability of default modeling-ifrs 9 challenges. Journal of Central Banking Theory and Practice, 8(1):209-223, 2019.

Yixiao Jiang. Semiparametric estimation of a corporate bond rating model. Econometrics, 9(2):23, 2021.

Gunter Loffler. An anatomy of rating through the cycle. Journal of Banking & Finance, 28(3):695-720, 2004.

Author Response

I appreciate your thoughtful comments and have made my best efforts to address these issues within the time that I have for this publication.  Anything that is a major extention will be highlighted as an enhancement for future research. 

Round 2

Reviewer 2 Report

The author has addressed my comments effectively.